# Deep Sedation for Dental Care Management in Healthy and Special Health Care Needs Children: A Retrospective Study

**DOI:** 10.3390/ijerph20043435

**Published:** 2023-02-15

**Authors:** Inmaculada Gómez-Ríos, Amparo Pérez-Silva, Clara Serna-Muñoz, Francisco Javier Ibáñez-López, Paula M. Periago-Bayonas, Antonio J. Ortiz-Ruiz

**Affiliations:** 1Department of Integrated Pediatric Dentistry, Biomedical Research Institute of Murcia, School of Dentistry, University of Murcia, 30008 Murcia, Spain; 2Scientific and Technical Research Area, Statistical Service, University of Murcia, 30008 Murcia, Spain; 3Agronomic Engineering Department, Universidad Politécnica de Cartagena (UPCT), Paseo Alfonso XIII, 48, 30203 Cartagena, Spain

**Keywords:** treatment under deep sedation, special health care needs children, quality of life, dental treatment, preventive program

## Abstract

Background: Very young children, and those with disabilities and extensive oral pathology, who cannot be treated in the dental chair, require deep sedation or general anesthesia for dental treatment. Objective: The aim of this study is to describe and compare the oral health status in healthy and SHCN children and the treatments performed under deep sedation on an outpatient basis with a minimal intervention approach, and their impact on quality of life. Methods: A retrospective study between 2006 and 2018 was made. A total of 230 medical records of healthy and SHCN children were included. The data extracted were age, sex, systemic health status, reason for sedation, oral health status before sedation, treatments administered during sedation, and follow-up. The quality of life after deep sedation of 85 children was studied through parental questionnaires. Descriptive and inferential analyses were made. Results: Of the 230 children, 47.4% were healthy and 52.6% were SHCN. The median age was 7.10 ± 3.40 years (5.04 ± 2.42 in healthy children and 8.95 ± 3.09 in SHCN children). The main reason for sedation was poor handling in the dental chair (99.5%). The most frequent pathologies were caries (90.9%) and pulp pathology (67.8%). Healthy children had more teeth affected by decay and with pulp involvement. Patients aged < 6 years received more pulpectomies and pulpotomies. After treatment, parents stated that children were more rested and less irascible, ate better, increased in weight, and had improved dental aesthetics. Conclusions: Differences in treatments carried out did not depend on the general health status or the failure rate but on age, with more pulp treatments in healthy children who were younger, and more extractions near to the age of physiological turnover in children with SHCN who were older. Intervention under deep sedation with a minimally invasive treatments approach met the expectations of parents and guardians, as it improved the children’s quality of life.

## 1. Introduction

The American Association of Pediatric Dentistry (AAPD) established, in 2020, that special health care needs (SHCN) include any physical, developmental, mental, sensory, behavioral, cognitive, or emotional impairment or any limiting condition that requires medical management, health intervention, and/or the use of specialized services or programs. The condition may be congenital, developmental, or acquired by illness, trauma, or environmental cause, and may impose limitations on performing daily self-care activities or substantial limitations to a major life activity. The health care of patients with special needs requires specialized training, increased awareness and care, and adaptation measures beyond usual [1,2].

Managing the behavior of patients with SHCN and very young children can be challenging due to anxiety and difficulty understanding the dental procedure. Thus, a surgical approach may be indicated in children with acute anxiety, lack of cognitive maturity, disability or limiting medical conditions, and/or extensive oral pathology [2,3,4].

The drawbacks of general anesthesia in children include their potential neurotoxicity. In general terms, when choosing this approach, the ASA classification of the patient [5,6], the length of the intervention, and the amount of pathology to be treated should be taken into account [7], although according to the AAPD, caution should be exercised in the case of children aged < 2 years [2,3].

Outpatient interventions performed under deep sedation have other advantages such as a lower cost, greater availability of appointments, shorter total working time, less interference in the environment of the patient [6,8], and the tendency to carry out more conservative treatments than when interventions are performed under general anesthesia in a hospital setting [9].

Although dental treatment under general anesthesia is not without risk [10] and involves higher health care costs [9], treatment of dental pathology on a single day in children is justified, as it is associated with a clear increase in the quality of life of children. To study the changes in the quality of life following the children’s dental treatment under general anesthesia and the impact on their families, questionnaires designed by the authors themselves, the Early Childhood Oral Health Impact Scale (ECOHIS), the Child Oral Health-Related Quality of Life (COHRQoL), and the Family Impact Scale (FIS), have been used mainly. Most systematic reviews conclude that this type of approach has a positive impact on families’ lives and an improvement in children’s quality of life [11,12,13].

Many dentists carrying out this type of surgery prefer to adopt aggressive therapeutic attitudes to avoid future reinterventions [14,15,16,17]. Thus, Tahmassebi, Achol, and Fayle (2014) [18], in a retrospective study of treatments carried out in healthy children and those with SHCN under general anesthesia, recorded more teeth extracted per child (3.8 temporary teeth and 0.55 permanent teeth) than filled (2.8 temporary teeth and 0.5 permanent teeth). In addition, healthy children received significantly more fillings than children with SHCN and more pulpotomies. The same trend was observed by Barbería et al. (2007) [19], who found the most frequent treatment was extractions (3.5 ± 3.1), which were especially aggressive in children with SHCN aged < 6 years (4.2 ± 3.2). König et al. (2020) [20] found that although the most frequent treatment was fillings (4.7 ± 3.1), the mean number of teeth extracted was also very high (3.7 ± 3.2).

Minimally invasive treatments individualized according to the risk of caries, together with prevention, are the basic pillars of the new approach to the treatment of caries in the international action protocols agreed by scientific associations [21,22] and which should serve as a guide on which to plan treatments, regardless of the systemic health status and the type of approach.

It is necessary to change the classical way of planning interventions under general anesthesia or deep sedation, especially in SHCN children, and to prioritize as much as possible the minimum intervention.

There are few published studies on the treatment of oral pathology in children (healthy and SHCN) in a single day on an outpatient basis with an individualized and minimally invasive approach. Therefore, the aim of this study was to describe and compare the oral health status in healthy and SHCN children and the treatments performed in these two groups of children under deep sedation on an outpatient basis with a minimal intervention approach, and their impact on quality of life.

## 2. Materials and Methods

This article was written according to the STROBE statement [23]. A retrospective study was conducted in children treated for oral pathology under deep sedation in a private clinic in Cartagena (Murcia, Spain).

Inclusion criteria were healthy children and those with SHCN, aged 2–18 years, treated under deep sedation, in the time period 2006–2018, whose medical records were correctly completed.

After discarding records not correctly completed, 230 clinical records were included. All patients or their parents gave signed informed consent and received an information sheet. The study was approved by the Research Ethics Committee of the University of Murcia (ID:2034/2018).

Anesthetic procedures were carried out by a team of anesthesiologists and nurses. The dental interventions were performed by the same operator. In the first visit, the medical history, the examination of the oral status, assessment of the degree of patient cooperation, and the demands of parents and/or guardians on the approach to the case were recorded. On all occasions, the possibility of administering treatment using behavioral management techniques in the dental chair or in the operating room was offered, explaining the pros and cons of each scenario. The patients who finally chose to be treated in the operating room attended a pre-anesthesia appointment with the anesthesiologist, who decided whether they were candidates for outpatient deep sedation.

On the day of the intervention, the patient fasted and, after a first phase of rapid induction with sevoflurane, a venous line was placed and intravenous propofol was administered and the airway was secured with a laryngeal mask. Post-treatment, the patient remained in the post-anesthetic recovery room until mobility was achieved, with blood oxygen saturation in normal parameters and recovery of the level of consciousness according to the bispectral index. Subsequently, a telephone follow-up was carried out on the same day and the day after the intervention. Within 15–30 days, the patient received a checkup at the dental clinic.

The information extracted from the medical records was:

(I) At the first visit:A.Demographic data: age and sex.B.Systemic health status, differentiating between healthy children and those with SHCN.C.Reason for sedation.D.Assessment of oral health status prior to the intervention:
−Hygiene habits. The child was considered to have a hygiene habit when regularly brushing a minimum of twice a day.−Plaque on visual inspection (Yes or No).−Calculus on visual inspection (Yes or No).−Caries and number of teeth affected. Caries was defined as the loss of enamel integrity (ICDAS 3, 4, 5).−Pulp involvement and number of teeth affected. ICDAS 6 lesions, night pain, radiolucent image on X-rays, and abscesses were considered pulp involvement.−Root debris on visual inspection and number.−Teeth lost to dental pathology on visual inspection (number).


(II) On the day of the intervention:A.Treatments carried out.
−Obturation.−Direct pulp protection.−Pulpotomy.−Pulpectomy.−Endodontics.−Apexification.−Calculus removal.−Scaling and root planing.−Application of fluoride.−Exodontias.
B.The number of teeth treated.

(III) Follow-up:A.Attended checkup (Yes or No)B.Plaque on visual inspection (Yes or No)C.Need for medication for oral pathology (Yes or No)D.Improved eating (Yes or No)

On the day of the intervention and the day of the review, the parents/guardians of 85 patients were administered a questionnaire consisting of 9 dichotomous answer questions (YES or NO) to study the impact of the oral status on children’s lives. The initial questionnaire questions were:Does the child have frequent pain in the mouth?Have they woken at night due to dental pain?Has this prevented them from carrying out their usual daily routine?Has the child taken medication due to dental problems?Does the child have difficulty eating meat?Does the child have difficulty eating cold food?Does the child have difficulty eating hot food?Does the child feel discomfort about the appearance of their teeth?When brushing the teeth, does the child complain of pain?

The questions of the review questionnaire at one month were:Does the child have difficulty eating meat?Does the child have difficulty eating cold food?Does the child have difficulty eating hot food?Does the child eat better after the intervention?Has the child gained weight during this time?Does the child feel discomfort when brushing their teeth?Is the child more rested and less irascible?Is the child happy with their teeth?Has treatment under deep sedation met your expectations?

### Statistical Analysis

All data were collected and stored in a database and statistically analyzed using R software version 3.6.0 [24]. A descriptive analysis was made of all study variables. Continuous quantitative variables were compared two by two using the *t*-test, *t*-test with Welch correction, or the Mann–Whitney test, depending on the assumptions of normality and homoscedasticity. To compare discrete quantitative variables between more than two groups, the Kruskal–Wallis test plus the Dwass–Steel–Critchlow–Fligner test was used. To establish the relationship between two categorical or discrete quantitative variables, contingency tables were made using Pearson’s χ^2^ or Fisher’s exact test, depending on whether the assumptions were met, and Cramer’s V test. The survey was studied by means of a general descriptive analysis of each of the questions and a descriptive analysis with inference by the sociodemographic variables of age, sex, and health condition. Statistical significance was established as *p* < 0.05.

## 3. Results

Of the 230 patients included, 85 were regular patients of the clinic and 145 (63.05%) were referred. Of those referred, 38 were from the health center, 2 from the educational center, and 105 from other professionals. The review appointment 15–30 days post-intervention was attended by 192 children, of whom 128 had recorded data on the presence of plaque before and after treatment, 136 on the need for medication, and 85 on the impact of oral health on the child’s life before and after deep sedation.

The mean age was 7.10 ± 3.40 years, with a median of 7 years and quartiles 25 and 75 of 4 and 9 years, respectively. The ages that contributed the highest number of children were 4 years (n = 31), 6 years (n = 29), 7 years (n = 25), 8 years (n = 25), and 9 years (n = 25), and 61.74% were male.

A total of 47.4% were healthy patients (n = 109). In children with SHCN (52.6%; n = 121), general developmental disorders and cerebral encephalopathies and palsy were the most common diseases, representing 34.2% of the total sample (Figure 1). In healthy children, the mean age was 5.04 ± 2.42 years, and in children with SHCN, 8.95 ± 3.09 years.

The main reason for sedation was poor management in the dental chair (99.5%). The main pathologies treated were caries (90.87%), trauma (6.08%), maintenance (0.86%), eruptive delay (0.86%), eruptive delay and gingivitis (0.43%), gingivitis (0.43%), and caries and periodontal disease (0.43%).

Of the total sample, 79.57% did not have dental hygiene habits at home, and 90.86% had dental plaque. Caries was present in 90.87% of patients, with 45.22% having 5–10 affected teeth. Pulp pathology was diagnosed in 67.83% of patients, and 40.86% had 1 or 2 teeth affected; a total of 13.91% had root remains and 4.34% had missing teeth (Table 1).

No significant differences in hygiene habits were detected between healthy children and children with SHCN (*p* = 0.36). The percentage of children with dental plaque was similar in the two groups. However, children with SHCN had a higher level of calculus than healthy children (52.89% vs. 7.33%, respectively; *p* < 0.001). Both groups had a high level of caries, around 90% (*p* = 0.26). Healthy children had more affected teeth than children with SHCN (7.49 ± 4. 68 vs. 6.13 ± 4.54; *p* < 0.05). Likewise, pulp involvement was higher in healthy children (78.90% vs. 57.85%; *p* = 0.001), with more affected teeth per child. Root remains, the total number of missing teeth, and the number of missing teeth per child were similar in the two groups (Table 1).

Healthy patients underwent more pulpectomies (*p* < 0.001). However, patients with SHCN had more endodontics (*p* < 0.05), dental cleanings (*p* < 0.001), fluoride applications (*p* < 0.001), and tooth extractions (*p* < 0.05) (Table 2).

The mean number of filled teeth and teeth with pulpectomies was significantly higher in healthy children, while the number of teeth extracted and the mean number of teeth receiving sealants was lower. When extractions due to the proximity of dental replacement were excluded from the analysis, the number of teeth extracted per pathology was similar in both groups (Table 3).

Of the total number of patients sedated, 81 were aged <6 years, 129 were aged 6–12 years, and 20 were aged >12 years. Patients aged <6 years received significantly more pulpectomies (*p* < 0.001) and pulpotomies (*p* = 0.037), fewer calculus removals (*p* < 0.001) and exodontias (*p* < 0.001), and a higher mean number of filled teeth and teeth with pulpectomy. More patients aged 6–12 years received fluoride (*p* = 0.003), had more exodontias (*p* < 0.001), and had a higher mean number of teeth extracted per child. Patients aged >12 years received significantly more endodontic treatments, calculus removals, and sealants, and less fluoride (Table 4 and Table 5).

The presence of plaque was recorded in 128 patients before and after the sedation. Of the 118 patients presenting plaque before the intervention, 112 continued to present it. Only 14.54% of healthy children and 10.96% of children with SHCN had no visible plaque after treatment.

A total of 49.26% of the 136 patients who registered the need for medication had required antibiotics, with or without anti-inflammatories/antipyretics for oral pathology before sedation. The most commonly used antibiotic was amoxicillin plus clavulanic acid. Patients also received amoxicillin, clindamycin, and the combination of metronidazole plus spiramycin. The most consumed analgesic was ibuprofen, followed by paracetamol and dexketoprofen. After sedation, only 23.81% of the 105 patients with a record of taking pharmacological treatment for oral pathology needed to ingest any medication. 

A total of 85 parents and/or guardians of patients aged 2–18 years completed a questionnaire on the impact of the oral status on their children’s lives: 75% were children receiving sedation for the first time. The results showed 20% had had frequent pain in the mouth before treatment, and 82.35% had night pain: 27.1% had ingested medication for oral problems. After treatment, 43.5% of parents found children more rested and less irascible, and 56.5% were able to eat better and, significantly, more hot foods (*p* = 0.016). Weight increases were reported by 25% of parents. Pre-intervention, 18.8% of children did not like their dental aesthetics and, after the intervention, 84.7% were happy with their teeth (*p* < 0.001). Treatment met the expectations of 95% of parents. The survey results were not influenced by age, sex, or whether children were healthy or had SHCN (Table 6 and Table 7).

## 4. Discussion

We studied the oral health status in 109 healthy children and 121 children with SHCN undergoing outpatient treatment under deep sedation and laryngeal mask. The lack of patient collaboration coincided with that described in other studies [15,24] and was the reason for 99.50% of sedations. No serious surgical or post-surgical complications were recorded, regardless of the ASA classification.

The mean age of children was 7.1 ± 3.4 years: 5.04 ± 2.42 in healthy children and 8.95 ± 3.09 in children with SHCN. The age of children included in comparable studies varies. Some have a similar mean age to ours [17,18,25], while in others, the age is lower [8,26,27]. It seems that the general health status determines the age: in healthy children, the age is lower [28], since healthy children learn to receive treatment in the dental chair as they mature, while children with SHCN have greater difficulty collaborating despite the passage of time [26]. In our region, there is a public oral health program in which children with mental, physical, or sensory disabilities, aged 6–14 years, have free access to oral treatments under anesthesia for deciduous and permanent teeth [29].

A total of 79.57% of the patients had no hygiene habits at home and 90.86% had abundant bacterial plaque at the first visit. At the ages of the children in our study, brushing with fluoride toothpaste (1000 to 1500 ppm) twice a day is essential as a caries prevention measure, as recommended by the WHO [30], together with parental supervision to ensure adequate oral hygiene. Olley et al. (2011) [25] found that 54% of children aged <7 years treated in the operating room brushed their teeth without parental supervision and Razeghi et al. (2020) [27] found similar results in 50% of a sample of children aged 2–5 years. A total of 90.87% of patients in our study had caries, with 45.22% having 5–10 teeth affected, and 67.83% had pulp involvement. The mean number of teeth with caries per child was 6.78 ± 4.65, similar to European studies (7.3 ± 4.3 [31] or 7.7 [32]), but lower than studies in Taiwan (15.16 in healthy patients; 15.21 in patients with SHCN) [26] and Iran (10.6 ± 4) [27]. 

Healthy children had significantly more caries and pulp involvement and more affected teeth at the first visit than children with SHCN. This difference was not observed by González De la Fuente (2015) [33]. In contrast, other authors observed a higher prevalence of caries in patients with mental disorders, who were institutionalized [34] or not [35], due to the lack of preventive programs aimed at this specific group.

Fillings were necessary in 91.73% of patients and exodontia in 38.70%. Most studies also identified dental restorations as the main treatment [14,20,31,36], as they give priority to restorative treatment over extractions to avoid oral dysfunction. However, some authors recorded exodontia as the most common treatment [37] or the second most common [20,36]. This attitude is intended to achieve more definitive results [14,15], forgetting the importance of temporary teeth in children’s physical, functional, and psychological development.

After fillings, preventive therapies were the second most frequent: calculus removal (59.13%), sealants (40.87%), and fluoride application (83.48%). Although most authors insist on the need to implement preventive programs, few studies prioritize this type of treatment [19,36]. In children with SHCN, preventive strategies should be a priority, as they may be at increased risk for oral diseases, which would further jeopardize the patient’s overall health. A dental home should be established before 12 months of age, along with an individualized oral hygiene program adapted to the child’s own disability, brushing supervised by parents/caregivers with fluoride paste twice a day, flossing, consumption of non-cariogenic diets except for special indications, in which case special care should be established to reduce the risk of caries, and use of sealants for primary and permanent teeth, among other measures [38].

Pulp treatments are rarely performed under general anesthesia or deep sedation [39]. In fact, Schnabl et al. (2020) [40] did not perform pulp protection, pulpotomy, or endodontic to reduce the intervention time and ensure the absence of long-term pain, even when they had not presented acute or chronic pain pre-intervention. In our study, 53.04% of patients underwent some type of pulp treatment, with pulpectomies at 33.91%. These treatments met the same quality requirements as those performed in the dental chair, so the success rate is the same as that found in other studies [15,41].

The mean number of teeth extracted in both groups was much lower than that reported by other studies [16,17,18,19,26,37,40], due to our more conservative approach. The number of exodontias carried out in patients with SHCN was significantly higher than that in healthy patients (0.64 vs. 1.35; *p* = 0.001), as shown by most published studies [16,17,26,37,40]. Many authors suggest that the higher number of exodontias in patients with SHCN is due to a worse oral health status [16,17]. However, it has been shown that both the level of plaque [19] and the number of teeth affected by caries [17,26,33] are independent of the general health status. In our sample, the difference was due to the fact that patients with SHCN, because of their age, 8.95 ± 3.09 years, were undergoing dental replacement and had difficulties in exfoliation due to a lack of chewing. In fact, the mean number of teeth extracted because of exfoliation was close to 0.15 in healthy patients and 0.54 in SHCN patients, and the mean number of teeth extracted due to disease in healthy children was 0.49 and 0.81 in children with SHCN, without significant differences. This increase in the extraction/restoration ratio due to the higher mean age of the sample was also observed by Tahmassebi, Achol, and Fayle (2014) [18].

Patients with SHCN have more exodontias to avoid the systemic risk caused by treatment failure [16,17,18,26,42]. Patients with SHCN in our sample were mainly those with general developmental disorders, encephalopathies, and cerebral palsy in which no vital organ is compromised if dental treatment fails. Dental treatments do not fail more in patients with SHCN than in healthy patients [19] and were not the main cause of subsequent interventions [18]. 

The healthy children in our study with a diagnosis of pulp involvement underwent more pulp treatments than children with SHCN, as recorded in other studies [15,18,26]. Pulpectomies were significantly more frequent in healthy children due to the lower age of this group and since this treatment is indicated exclusively in the temporary dentition.

Carrying out all the treatment in a single day significantly means an improvement in the quality of life [12,13], regardless of whether treatment is carried out under general anesthesia or sedation [7]. Although the survey conducted in our study is not validated, it “reflects people’s comfort in eating, sleeping and interacting with others; their self-esteem and their satisfaction with their oral health” combining symptoms and life experiences and representing the subjective perspective [12]. Thus, after treatment under sedation, 43.5% of parents found their children were more rested and less irascible, with weight gain in 25% of cases. Ferrazzano et al. (2019) [43] found an increase in height in 44% and in weight and the body mass index in 55% of their sample of healthy children.

This study had some limitations. It was an observational retrospective study, and prospective studies should be carried out to assess the improvement in the quality of life of patients undergoing deep sedation or general anesthesia with validated quality of life surveys. Validated oral quality of life scales such as ECOHIS (Early Childhood Oral Health Impact Scale) [44] and COHRQoL (Child Oral Health-Related Quality of Life) [13] should also be used to compare our results with other studies.

## 5. Conclusions

Healthy children had more caries lesions and higher frequency of pulp involvement than SHCN children. Differences in treatment for both groups were not determined by the systemic health status or a different failure rate, but were determined by the different mean ages, resulting in more pulp treatments in healthy children, aged 5.04 ± 2.42, and more extractions of teeth close to physiological turnover in SHCN children, with 8.95 ± 3.09 years.

Intervention under deep sedation is a safe and necessary type of approach, both for healthy and SHCN children, where minimally invasive treatments can be carried out with highly satisfactory and predictable results. It met the expectations of parents and guardians, as it improved the children’s quality of life.

## Figures and Tables

**Figure 1 ijerph-20-03435-f001:**
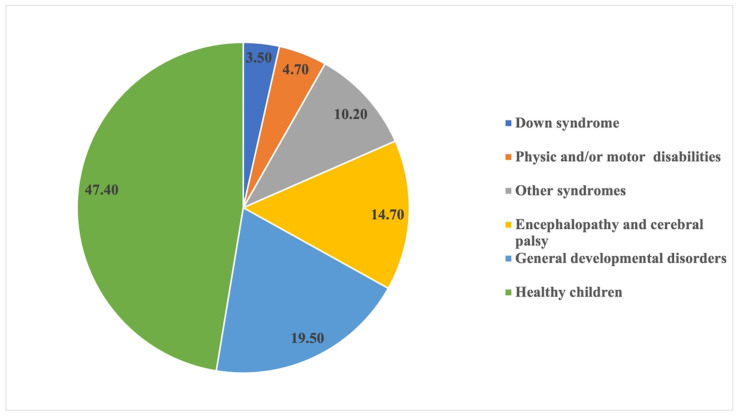
Distribution of healthy and special health care needs children.

**Table 1 ijerph-20-03435-t001:** Initial oral health status in healthy and special health care needs children.

	Total (100%)	Healthy (47.4%)	SHCN (52.6%)	*p*-Value
Tooth brushing habit	20.43%	17.43%	23.14%	=0.36 ^†^
Plaque	90.86%	93.57%	89.25%	=0.35 ^†^
Calculus	31.30%	7.33%	52.89%	<0.001 ^†^
Caries	90.87%	93.57%	88.43%	=0.26 ^†^
Pulp involvement	67.83%	78.90%	57.85%	=0.001 ^†^
Root remains	13.91%	10.09%	17.36%	=0.16 ^†^
Missing Teeth	4.34%	4.58%	4.13%	=1 *
Number of teeth with caries per child (mean ± SD)	6.78 ± 4.65	7.49 ± 4.68	6.13 ± 4.54	<0.05 ^‡^
Number of teeth with pulp involvement per child (mean ± SD)	1.84 ± 2.04	2.25 ± 2.01	1.47 ± 2.00	=0.004 ^‡^
Number of missing teeth per child (mean ± SD)	0.10 ± 0.84	0.045 ± 0.21	0.16 ± 1.14	=0.005 ^‡^

SHCN: special health care needs children. *p*-value healthy vs. SHCN. * Fisher’s exact test. ^†^ Pearson’s χ^2^ test. ^‡^ Mann–Whitney U test.

**Table 2 ijerph-20-03435-t002:** Number of patients who received the different treatments.

	Total (100%)		Healthy (47.4%)		SHCN (52.6%)		
	%	n	%	N	%	n	*p*-Value
Fillings	91.73	211	95.41	104	88.43	107	>0.05 ^†^
Direct Pulp capping	1.30	3	0.91	1	1.65	2	>0.05 *
Pulpectomy	33.91	78	55.96	61	14.05	17	<0.001 ^†^
Pulpotomy	13.04	30	16.51	18	9.91	12	>0.05 ^†^
Endodontic	13.04	30	6.42	7	19.00	23	<0.05 ^†^
MTA apexfication	1.30	3	0.91	1	1.65	2	>0.05 *
Calculus removal	59.13	136	29.36	32	85.95	104	<0.001 ^†^
Fissure sealants	40.87	94	36.70	40	44.63	54	>0.05 ^†^
Scaling and root planing	0.86	2	0	0	1.65	2	>0.05 *
Fluoride application	83.48	192	73.39	80	92.56	112	<0.001 ^†^
Exodontias	38.7	89	31.19	34	45.45	55	<0.05 ^†^
Exodontias due to disease	30	69	25.69	28	33.88	41	>0.05 ^†^

MTA: Mineral Trioxide Aggregate. SHCN: special health care needs children. *p*-value healthy vs. SHCN. * Fisher’s exact test. ^†^ Pearson’s χ^2^ test.

**Table 3 ijerph-20-03435-t003:** Mean number of treatments in healthy and special health care needs children.

Treatment	Healthy (47.4%)	SHCN (52.6%)	*p*-Value
Fillings	6.85 ± 4.55	4.95 ± 3.72	<0.05 ^¶^
Direct pulp capping	0.027 ± 0.28	0.016 ± 0.12	>0.05 ^¶^
Pulpectomy	1.26 ± 1.61	0.19 ± 0.52	<0.001 ^§^
Pulpotomy	0.32 ± 0.84	0.20 ± 0.75	>0.05 ^¶^
Endodontics	0.15 ± 0.82	0.23 ± 0.53	>0.05 ^§^
MTA apexification	0.01 ± 0.09	0.016 ± 0.13	>0.05 ^¶^
Fissure sealants	0.88 ± 1.54	1.80 ± 2.75	=0.002 ^¶^
Extractions	0.64 ± 1.36	1.35 ± 2.04	=0.002 ^§^
Extracted due to disease	0.49 ± 1.19	0.81 ± 1.48	>0.05 ^¶^

MTA: Mineral Trioxide Aggregate. SHCN: special health care needs children. *p*-value healthy vs. SHCN. ^¶^
*t*-test. ^§^
*t*-test with Welch correction.

**Table 4 ijerph-20-03435-t004:** Type of treatment according to age.

	<6 Years (n = 81)	6–12 Years (n = 129)	>12 Years (n = 20)	*p*-Value + Cramer’s V
Fillings	93.80%	90.69%	90.00%	>0.05 *
Direct pulp capping	1.23%	0.77%	5.00%	>0.05 *
Pulpectomy	61.85%	27.9%	0.00%	<0.001 ^†^ (V = 0.32)
Pulpotomy	19.70%	10.85%	0.00%	=0.037 * (V = 0.17)
Endodontic	0.00%	15.50%	50.00%	<0.001 * (V = 0.379)
MTA apexfication	0.00%	2.32%	0.00%	>0.05 *
Calculus removal	14.81%	82.17%	85.00%	<0.001 ^†^ (V = 0.666)
Scaling and root planing	0.00%	0.00%	10.00%	=0.006 * (V = 0.312)
Fissure sealants	32.09%	48.06%	30.00%	=0.025 ^†^ (V = 0.179)
Fluoride application	76.54%	90.69%	65.00%	=0.003 * (V = 0.242)
Exodontias	18.51%	51.16%	40.00%	<0.001 ^†^ (V = 0.309)

MTA: Mineral Trioxide Aggregate. * Fisher’s exact test. ^†^ Pearson’s χ^2^ test.

**Table 5 ijerph-20-03435-t005:** Mean number of teeth treated per child according to age.

Treatments	<6 Years	6–12 Years	>12 Years	*p*-Value
Fillings	6.88 ± 4.41	5.29 ± 3.53 ^#^	6.05 ± 6.53	<0.05
Direct pulp capping	0.04 ± 0.33	0.01 ± 0.09	0.05 ± 0.22	=0.305
Pulpectomy	1.19 ± 1.64	0.50 ± 1.01 ^#^	0.00 ± 0.00 ^#@^	<0.05
Pulpotomy	0.40 ± 0.89	0.22 ± 0.78	0.00 ± 0.00	=0.06
Endodontics	0.00 ± 0.00	0.19 ± 0.49 ^#^	0.95 ± 1.82 ^#@^	<0.05
MTA apexification	0.00 ± 0.00	0.02 ± 0.15	0.00 ± 0.00	=0.306
Fissure sealant	0.85 ± 1.57	1.60 ±2.31 ^#^	1.90 ± 3.99	<0.05
Exodontias	0.30 ± 0.73	1.50 ± 2.13 ^#^	0.85 ± 1.35	<0.05

MTA: Mineral Trioxide Aggregate. Values expressed as mean ± SD. Kruskal–Wallis test + Dwass–Steel–Critchlow–Fligner test. ^#^: *p* value vs. <6 years. ^@^: *p* value vs. 6–12 years.

**Table 6 ijerph-20-03435-t006:** Results of questionnaire on children’s quality of life pre-treatment made by the parents/guardians of 85 patients.

	Sex (Male/Female) N (57/28)	Age (2–6/7–11/12–18) N (29/40/16)	Health Condition (Healthy/SHCN) N (22/63)
1. Does the child have frequent pain in the mouth?	14/3 ^†^ (*p* = 0.25)	5/10/2 * (*p* = 0.62)	2/15 * (*p* = 0.22)
2. Have they woken at night due to dental pain?	12/2 * (*p* = 0.21)	5/7/2 * (*p* = 1.00)	5/9 * (*p* = 0.32)
3. Has this prevented them from carrying out their usual daily routine?	9/1 * (*p* = 0.16)	5/4/1 * (*p* = 0.54)	4/6 * (*p* = 0.27)
4. Has the child taken medication due to dental problems?	19/4 ^†^ (*p* = 0.14)	9/11/2 * (*p* = 0.50)	7/16 ^†^ (*p* = 0.64)
5. Does the child have difficulty eating meat?	20/8 ^†^ (*p* = 0.85)	9/15/3 * (*p* = 0.53)	6/22 ^†^ (*p* = 0.72)
6. Does the child have difficulty eating cold food?	14/2 ^†^ (*p* = 0.11)	5/9/2 * (*p* = 0.88)	3/13 * (*p* = 0.75)
7. Does the child have difficulty eating hot food?	12/1 * (*p* = 0.05)	4/7/2 * (*p* = 1.00)	3/10 * (*p* = 1.00)
8. Does the child feel discomfort about the appearance of their teeth?	9/2 * (*p* = 0.49)	5/5/1 * (*p* = 0.71)	5/6 * (*p* = 0.13)
9. When brushing the teeth, does the child complain of pain?	18/6 ^†^ (*p* = 0.53)	10/11/3 * (*p* = 0.62)	4/20 ^†^ (*p* = 0.40)

The answers were dichotomous (Yes/No). The table reflects only the positive answers. SHCN: special health care needs children. * Fisher’s exact test. ^†^ Pearson’s χ^2^ test.

**Table 7 ijerph-20-03435-t007:** Results of questionnaire on children’s quality of life post-treatment made by the parents/guardians of 85 patients.

	Sex (Male/Female) N (57/28)	Age (2–6/7–11/12–18) N (29/40/16)	Health Condition (Healthy/SHCN) N (22/63)
1. Does the child have difficulty eating meat?	15/6 ^†^ (*p* = 1.00)	4/13/3 * (*p* = 0.62)	3/18 ^†^ (*p* = 0.23)
2. Does the child have difficulty eating cold food?	10/4 * (*p* = 1.00)	4/10/0 * (*p* = 0.08)	3/11 * (*p* = 1.00)
3. Does the child have difficulty eating hot food?	3/1 * (*p* = 1.00)	1/3/0 * (*p* = 0.65)	2/2 * (*p* = 0.26)
4. Does the child eat better after the intervention?	36/12 ^†^ (*p* = 0.11)	18/22/7 ^†^ (*p* = 0.75)	11/37 ^†^ (*p* = 0.68)
5. Has the child gained weight during this time?	15/7 ^†^ (*p* = 1.00)	9/11/2 * (*p* = 0.52)	7/15 ^†^ (*p* = 0.62)
6. Does the child feel discomfort when brushing their teeth?	9/8 ^†^ (*p* = 0.19)	5/9/3 * (*p* = 0.93)	3/14 * (*p* = 0.54)
7. Is the child more rested and less irascible?	27/10 ^†^ (*p* = 0.43)	13/15/8 ^†^ (*p* = 0.44)	9/28 ^†^ (*p* = 1.00)
8. Is the child happy with their teeth?	51/21 * (*p* = 0.58)	26/31/14 * (*p* = 0.54)	21/51 * (*p* = 0.57)
9. Has treatment under deep sedation met your expectations?	55/26 * (*p* = 1.00)	27/37/15 * (*p* = 1.00)	21/60 * (*p* = 1.00)

The answers were dichotomous (Yes/No). The table reflects only the positive answers. SHCN: special health care needs children. * Fisher’s exact test. ^†^ Pearson’s χ^2^ test.

## Data Availability

The datasets used for the current study are available from the corresponding author upon reasonable request.

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
