# Peer review of "Deep Sedation for Dental Care Management in Healthy and Special Health Care Needs Children: A Retrospective Study"

_ijerph, 2023, doi:10.3390/ijerph20043435_

Round 1

Reviewer 1 Report

MDPI

A review for the international journal of environmental research and public health:

This is an interesting paper that addresses oral health and treatments carried out under deep sedation in impact on the quality of life of healthy children and those with special health care needs. Some issues should be addressed before a proper evaluation of the manuscript. 

General comment: the manuscript requires English editing. 

Introduction: 

Elaborate on the associated increase in quality of life in P 2 L68, please.

While the introduction is comprehensive and covers the base concepts and trends related to the research's main topic, it lacks the logical sequence that leads to the rationale of conducting the studies. Please add a paragraph that clearly states the gaps in knowledge and rationale for conducting the study. 

Methods:

P2 L93: Please cite the STROBE statement. 

P2L 96: Were there any inclusion criteria that led to reaching this particular number? Or were these all the completed files? 

P3 L129: I would change tartar to “calculus”. Consider changing, please.

P5 L156: when exactly have these questionnaires were administered to the parents?

L188: I would change qualitative to “categorical”

Results:

Figure 1 does not include the light blue colour. Add, please.

Table 1: Please clarify what is meant by “teeth lacking.

Table 1: please add the frequencies for each status as well.

Table 2: I would change general to “total”

Table 4: check the p values direction, please.

P10 L298-299: the authors did not report the data, how exactly was this tested?

Discussion: 

“The different mean age of the two study cohorts may have conditioned different treatment plans” I elaborate on this, please. 

Another limitation that was not addressed is related to the fact that the impact on quality of life was not measured using a validated tool and it was only measured on a sub-sample. 

The first sentence f the conclusion should be rewritten to reflect the sample of the study. 

Author Response

All changes are in word file Review 1

Reviewer 2 Report

Dear authors,

this article is quite interesting, however it would be better if you performed some validated Quality of life questionnaire - the value of your investigation would significantly improve. This being said, I just have some minor remarks regarding the manuscript.

In line 42 you have too many "and"-s, re-write.

You did not mention quality of life in the Introduction section - how is it assessed, references to possible previous similar studies, etc.

Line 144 - the correct term is apexification, just as you put it in tables.

I have a problem with fluoride application as treatment, I am afraid it is nothing worth mentioning in this context. Why would you put a child in sedation for that? It is a very common preventive measure which is not limited exclusively to dental professionals.

What does it mean that children did not have hygiene habits at home? They did not brush or no one instructed them? Or they brushed, but without supervision? In Discussion you should also add some remarks about present day recommendations, you merely compare results, and Discussion should include some insights, guidelines, comments, opinions, clinical recommendations... Discussion should be altered.

And finally, Conclusions are too long - please limit it to 2-3 sentences.

Author Response

All changes are in word file review 2

Reviewer 3 Report

Dear authors,

thank you for your well arranged article. There are only 2 minor quires that need to be addressed.

1.       In figure 1 please change “sindrome” to “syndrome”.

2.       Table 1: What is meant by lacking teeth? Did you mean missing teeth? Please correct.

3.       You might also condense some of the information provided in the introduction

Author Response

All changes are in word file review 3

Round 2

Reviewer 1 Report

The manuscript has significantly improved. The authors successfully addressed all my comments. I have no further comments to add.